# Polyhydroxyalkanoate/Antifungal Polyene Formulations with Monomeric Hydroxyalkanoic Acids for Improved Antifungal Efficiency

**DOI:** 10.3390/antibiotics10060737

**Published:** 2021-06-18

**Authors:** Marina Pekmezovic, Melina Kalagasidis Krusic, Ivana Malagurski, Jelena Milovanovic, Karolina Stępień, Maciej Guzik, Romina Charifou, Ramesh Babu, Kevin O’Connor, Jasmina Nikodinovic-Runic

**Affiliations:** 1Institute of Molecular Genetics and Genetic Engineering, University of Belgrade, Vojvode Stepe 444a, 11221 Belgrade, Serbia; marinapekmezovic@gmail.com (M.P.); Ivana.malagurski@gmail.com (I.M.); jelenaradivojevic@imgge.bg.ac.rs (J.M.); 2Leibniz Institute for Natural Product Research and Infection Biology, Department of Microbial Pathogenicity Mechanisms, Hans Knoell Institute, Beutenberstrasse 11a, 07745 Jena, Germany; 3Faculty of Technology and Metallurgy, University of Belgrade, Karnegijeva 4, 11000 Belgrade, Serbia; meli@tmf.bg.ac.rs; 4Centre for Preclinical Research and Technology, Department of Pharmaceutical Microbiology, Faculty of Pharmacy, Medical University of Warsaw, Banacha 1B, 02-097 Warsaw, Poland; kstepien@wum.edu.pl; 5Jerzy Haber Institute of Catalysis and Surface Chemistry Polish Academy of Sciences, Niezapominajek 8, 30-239 Krakow, Poland; maciej.guzik@ikifp.edu.pl; 6AMBER Centre, CRANN Institute, School of Chemistry, Trinity College Dublin, D2 Dublin, Ireland; CHARIFOR@tcd.ie (R.C.); babup@tcd.ie (R.B.); 7BiOrbic Bioeconomy SFI Research Centre, University College Dublin, Belfield, D4 Dublin 4, Ireland; kevin.oconnor@ucd.ie; 8School of Biomolecular and Biomedical Sciences, University College Dublin, Belfield, D4 Dublin 4, Ireland; 9Centre for Synthesis and Chemical Biology, University College Dublin, Belfield, D4 Dublin 4, Ireland

**Keywords:** polyhydroxyalkanoate, film, antifungal, 3-hydroxydecanoic acid, polyene, nystatin, amphotericin B, *Candida*

## Abstract

Novel biodegradable and biocompatible formulations of “old” but “gold” drugs such as nystatin (Nys) and amphotericin B (AmB) were made using a biopolymer as a matrix. Medium chain length polyhydroxyalkanoates (mcl-PHA) were used to formulate both polyenes (Nys and AmB) in the form of films (~50 µm). Thermal properties and stability of the materials were not significantly altered by the incorporation of polyenes in mcl-PHA, but polyene containing materials were more hydrophobic. These formulations were tested in vitro against a panel of pathogenic fungi and for antibiofilm properties. The films containing 0.1 to 2 weight % polyenes showed good activity and sustained polyene release for up to 4 days. A PHA monomer, namely 3-hydroxydecanoic acid (C10-OH), was added to the films to achieve an enhanced synergistic effect with polyenes against fungal growth. Mcl-PHA based polyene formulations showed excellent growth inhibitory activity against both *Candida* yeasts (*C. albicans* ATCC 1023, *C. albicans* SC5314 (ATCC MYA-2876), *C. parapsilosis* ATCC 22019) and filamentous fungi (*Aspergillus fumigatus* ATCC 13073; *Trichophyton mentagrophytes* ATCC 9533, *Microsporum gypseum* ATCC 24102). All antifungal PHA film preparations prevented the formation of a *C. albicans* biofilm, while they were not efficient in eradication of mature biofilms, rendering them suitable for the transdermal application or as coatings of implants.

## 1. Introduction

Polyhydroxyalkanoates (PHAs) as biodegradable and bio-based materials have gained much of interest lately in a view of the global plastic waste crisis [1,2,3]; however, they also possess unique material properties to meet biomedical application-specific requirements [4,5,6]. PHAs are family of natural bacterial biopolyesters composed of a wide variety of hydroxyalkanoates monomers giving them tunable mechanical properties from brittle rigid plastics to gummy elastomers [7]. Due to their biocompatibility and versatility PHAs have been developed into medical implants, artificial organ constructs, cell scaffolds for tissue replacement, repair and regeneration, wound dressings and drug delivery carriers [8,9,10,11].

The most common and investigated PHA is poly-3-hydroxybutyrate (PHB). The degradation products of PHB were found to be not toxic and to not cause inflammatory responses both in vitro and in vivo [12,13,14]. Medium chain length PHAs (mcl-PHAs) are also attracting interest due to unique properties and specific monomer composition [15,16,17]. Furthermore, mcl-PHA-derived 3-hydroxyoctanoic acid and its derivatives were shown to have antimicrobial properties [18], while 3-hydroxydecanoic acid was used to enhance the anticancer properties of peptides [19,20]. Nevertheless, antifungal properties of 3-hydroxyacids have not been assessed and formulations of mcl-PHA for anti-fungal agent delivery have not been made or evaluated.

Fungal infections, usually caused by opportunistic pathogens, affect nearly a billion people worldwide [21]. The majority of those infections are superficial, localized on the mucosa, skin, nail and hair. However, more than 150 million people suffer from the invasive form of infection, which is often fatal with an estimated 1.7 million deaths per year [21,22]. *Candida* spp. are the most common cause of invasive fungal infections [23,24]. *Candida* biofilms are more virulent than their planktonic counterpart, cause chronic infections and exhibit strong resistance to antifungal treatments [25,26,27]. *Candida* biofilms can be formed on a wide range of both biotic and abiotic surfaces. The incidence of biofilm-associated candidemia in patients with implants such as catheters, dental implants and voice prostheses is rising [28,29,30]. Therefore, development of antimicrobial and biofilm inhibitive materials or coatings are important for the management of recalcitrant *Candida* biofilm associated infections.

Introduced in 1950s, polyenes represent the oldest family of antifungal drugs [31,32]. Despite their potential toxicity, polyenes remain useful in the treatment of invasive fungal infections because of their broad-spectrum activity, low rates of resistance and established clinical record, particularly in immunocompromised patients. Nystatin is a polyene antifungal drug that is of particular interest because it exhibits remarkable action against a wide range of pathogenic and non-pathogenic yeast and fungi and it is used in severely immune compromised patients and neonates [33,34,35]. While different formulations of nystatin have been developed [36], the topical formulation of nystatin remains the most widespread in clinical use [37]. Amphotericin B deoxycholate has been the “gold standard” treatment for invasive fungal infections for over 50 years [31,38,39]. Driven to improve on the renal toxicity of amphotericin B deoxycholate, extensive pharmaceutical research has led to the development of several new amphotericin B formulations. Compared with amphotericin B deoxycholate, the lipid formulations of amphotericin B (amphotericin B lipid complex, amphotericin B colloidal dispersion and liposomal amphotericin B) have distinct advantages in improved drug safety. However, the lipid formulations of amphotericin B are significantly more expensive than amphotericin B deoxycholate [40]. More recently, their activity has been evaluated against growing problem of *Candida* biofilms [41,42].

On one side, there is a clear need for biomaterials that can be used for safe delivery of somewhat toxic polyenes, while these materials should also stop the attachment, proliferation and formation of device-associated biofilms. Towards that goal, in this study biocompatible and biodegradable mcl-PHA films were prepared in order to investigate their potential as carriers for nystatin and amphotericin B. The obtained samples were characterized and evaluated in terms of their antifungal activity against the range of different human fungal pathogens and the potential to inhibit *Candida* spp. biofilm formation.

## 2. Results

PHAs are composed of repeating units of 3-hydroxyalaknaoic acids (R3HAs) which have been reported to have moderate anti-bacterial activity [18,43]. However, they have not been investigated for their anti-fungal activity. As part of the study of adding polyenes to PHA we also wished to examine the anti-fungal activity of R3HAs to see if there could be an opportunity to add these in with polyenes in the biopolymer matrix to enhance the anti-fungal effect.

### 2.1. Antifungal Properties of PHA Monomers, Nystatin and Amphothericin B

Minimum inhibitory concentrations (MICs) of (*R*)-3-hydroxyoctanoic (C8-OH) and (*R*)-3-hydroxydecanoic (C10-OH) acids, as mcl-PHA monomers, along that of nystatin (Nys) and amphotericin B (AmB) have been evaluated in the standard broth microdilution assay and their cytotoxicity against a healthy human fibroblast cell line was determined (Table 1). AmB was more efficient against the range of tested fungi in comparison to Nys, with MIC values 2–16-fold lower than Nys. However, its IC_50_ value for human fibroblast cells (MRC-5) was also over 4-fold lower compared to Nys (Table 1). Both mcl-PHA monomers showed moderate to poor activity against fungal isolates (two or three orders of magnitude higher MICs in comparison to the polyenes), with C10-OH being more active than C8–OH especially in the case of *C. parapsilosis* (Table 1). Interestingly, when C10-OH monomer was combined with AmB some positive correlation was observed. Low amounts of C10-OH (6.25–15.6 µg/mL which is up to 80-fold lower in comparison to MIC values) were able to reduce the MIC of AmB against all fungal species, except *T. mentagrophytes* (Table 1). These combinations of C10-OH/AmB showed comparable or lower cytotoxicity to C10-OH for human fibroblast (MRC-5) cells. This synergistic effect was not observed in the case of C8-OH monomer with either of the polyenes, and only an additive effect was observed when using C10-OH and Nys (data not shown).

### 2.2. Mcl-PHA/Polyene Films Morphology and Microstructure

Novel mcl-PHA/polyene films were successfully produced by solvent casting. In order to find optimal formulations in terms of antifungal activity and film properties, the mass of incorporated polyenes and film thickness, were varied (Table 2). 

Prepared mcl-PHA-based films (a copolymer of C8-OH and C10-OH, polyhydroxyoctanoate-co-decanoate (P3(HO-co-DO) in 1:1 ratio) were flexible with the average thickness of 50 ± 0.01 µm. The unmodified film (mcl-PHA-control) was transparent while films loaded with nystatin (NYS) and amphotericin B (AMB) were yellow and non-transparent (Figure 1A). Nystatin and amphotericin B encapsulation efficacy was 100% and the coloration of the prepared films was dependent on the drug concentration (Figure 1A). The distribution of the polyenes through the film was uniform as per macroscopic observations and no observable differences were noted under scanning electron microscope (Figure 1B). Preliminary antimicrobial tests highlighted samples NYS-9, NYS-17, AMB and AMB-M as the most promising ones (i.e. showing the largest inhibition zones), so these formulations were chosen for further characterization. The zones of inhibited fungal growth were apparent in the films at 2% wt of nystatin (NYS-17) and as low as 0.1% wt of AmB (AMB). Stable and uniform films containing more than 1% wt AmB could not be obtained using solvent casting method.

The neat mcl-PHA sample was semi-crystalline with a melting point of 49.31 °C (Appendix A, Appendix A). There was no change in thermal properties of the polymer with addition of Nys. However, a slight decrease of 5 °C in the melting temperature was noted for the samples loaded with AmB. The addition of polyene molecules to mcl-PHA increased the thermal decomposition of modified polymer marginally. The polyene-containing films were, generally, more hydrophobic than the neat mcl-PHA film (Appendix A), as an increase between 7 and 12 °C in water contact angle values was observed.

### 2.3. Polyene Release Kinetics from Mcl-PHA/Polyene Films

Once it was confirmed that mcl-PHA/polyene films showed antifungal activity, release of Nys and AmB from the samples was investigated. Having in mind the potential application of the films as a wound dressing against fungal infection, the release of polyenes was carried out in simulated sweat at pH 5.5 and at 37 °C. UV/VIS spectroscopy revealed that both Nys and AmB were successfully released from the films over the period of 4 days (Figure 2). Overall, 66.9% and 61.7% of AmB and Nys was released at 37 °C, respectively, in this time period, and no visible material deterioration was detected in the simulated sweat.

### 2.4. In Vitro Antifungal Activity of Polyenes Immbolised in mcl-PHA

Selected film formulations of both polyenes NYS-9 and NYS-17, as well as AMB and AMB-M showed excellent activity against all tested fungal strains in direct contact using an agar diffusion assay (Figure 3). More importantly, this activity was stable and slightly improved, even after 30 days of storage at 4 °C. Formulations containing a 10-fold higher amount of Nys (NYS17 in comparison to NYS9), were more efficient in fungal growth inhibition as reflected in a 2-fold larger zone of inhibition diameters (Figure 3A). In agreement with the MIC values, reported in Table 1, films loaded with AmB exhibited higher activity in comparison to NYS-loaded films (Table 1, Figure 3B).

In addition to the activity on the agar plates (Figure 4A), AMB-M films containing 1 wt % of AmB and 0.04 wt % of C10-OH monomer, were also tested in the liquid SAB media and showed the complete growth inhibition of *C. albicans* and *T. mentagrophytes* (Figure 4B), while the growth of *M. gypseum* was 1.2-fold reduced compared to the control of fungal growth in media only. Neat mcl-PHA film caused no inhibition of fungal growth (results not shown).

### 2.5. Inhibition of C. albicans Biofilm Formation

A biofilm inhibition test was conducted using a confirmed biofilm producer clinically isolated strain *Candida albicans* 329UCK. All tested polyene containing samples exhibited antifungal activity against the isolate in the agar screen (Appendix A).

In the liquid culture, after 48 h, the NYS-9, AMB and AMB-M films had significantly reduced *C. albicans* 329UCK growth. The NYS-17 film caused complete growth inhibition of yeast and prevented biofilm formation. Results are expressed as a log_10_ CFU sample^−1^ of cells adhered to materials and presented in Figure 5A. The results demonstrate high log-reductions up to 5 log_10_ (CFU/sample) of attachment of the yeast to the AMB films (4.67 × 10^1^ CFU/sample) and AMBM films (8 × 10^1^ CFU/sample). The NYS-9 film inhibited the yeast adhesion about 1.5 log_10_ reduction (4.09 × 10^5^ CFU/sample) in comparison to control mcl-PHA film (1.6 × 10^7^ CFU/sample). Microscopic analyses of these mcl-PHA films supported the adhesion assessment (Figure 5B). At AMB and AMBA surfaces the budding and aggregates of yeast cells were observed whereas at NYS-17 film only limited number of individual yeast cells appeared. The more developed *Candida* biofilm was observed at the control mcl-PHA surfaces and also started to develop at NYS-9 film, within this time frame. In both cases, biofilms contained the hyphae, pseudo-hyphae and budding yeast forms (Figure 5B).

Finally, to evaluate the ability of the polyene containing samples to eradicate formed *Candida* spp. biofilms, the micropore method was used whereby polyene-containing films were placed on the pre-formed mature *Candida* biofilms and the residual biofilm was evaluated. It was observed that none of the tested samples was able to eradicate 48 hour-old biofilms (Figure 6). *Candida* spp. growth was observed at every replica of the neat mcl-PHA and mcl-PHA/polyene films along with streaking from the filters of their imprints (data not shown). The biomass of NYS-17 film appeared limited, whereas the other imprints were comparable. The filter imprints of *Candida* spp. growth were proportional for every mcl-PHA-polyenes film.

## 3. Discussion

(*R*)-3-hydroxyalkanoic acids were produced biotechnologically from the mcl-PHA as a starting material [20,44]. They have been recognized as valuable synthons and molecules with bioactive properties [18,45]. (*R*)-3-hydroxyoctanoic acid (C8-OH) was shown to inhibit wide range of bacterial and two fungal strains namely *C. albicans* and *M. gypseum* with moderate efficiency and MIC values between 500 and 1000 µg/mL [18]. This finding was confirmed within the current study; however, we have observed that (*R*)-3-hydroxydecanoic acid (C10-OH) was more potent in comparison to C8-OH against the range of fungal strains (Table 1). More importantly, C10-OH potentiated the effect of AmB against the range of fungi including difficult to treat *A. fumigatus* and *M. gypseum*, and these combinations showed markedly reduced toxicity of the AmB when combined with this specific monomer. This is of special importance, bearing in mind that *A. fumigatus* is an opportunistic fungus that causes a severe and often fatal (mortality rate 50–95%) invasive aspergillosis [46,47]. New or improved treatment options for this pathogen that are intrinsically resistant to fluconazole [48] and capable of biofilm formation [47,49] are urgently needed. Bearing in mind that human and animal dermatophytosis worldwide are on the rise and are quite difficult to treat and eradicate [50,51,52], the observed increased sensitivity to AmB of dermatophytes, such as *M. gypseum* achieved in this study can be further explored. Stronger synergistic activity of liposomal AmB and antibacterial agent colistin has been previously reported resulting in up to 8-fold and 16-fold MIC reduction of liposomal AmB for *Candida* spp. and *Aspergillus* spp. [53]. A combination strategy has been applied with some azoles and other bioactive molecules such as 4-(5-methyl-1,3,4-thiadiazole-2-yl) benzene-1,3-diol in an attempt to minimize toxicity of AmB by reducing its dose [54,55]. Data from this study are encouraging in terms of further optimization of the formulation and possibly developing liposome-like PHA micelles for delivery of polyenes and other bioactive agents.

AmB has been widely used in the treatment of systemic fungal infections, but its therapeutic effectiveness is hampered by dose-limiting toxicity in vivo [56]. To date, it has not been used as topical antifungal agent [57]. Transdermal and topical delivery of drugs in general provides advantages over conventional oral administration in terms of convenience, improved patient compliance and elimination of hepatic first pass effect. Therefore, mcl-PHA formulations with reduced toxicity achieved in this study may widen the spectrum of AmB applications. On the other side, topical applications of nystatin are widespread; however, the sustained release films are scarce and usually based on synthetic polymers [58,59].

Antifungal films using different polymer matrices and antifungal agents are mostly developed as an effort to produce active food packing films [60,61,62,63]. Given that these films or coatings are usually edible and bio-degradable, they are usually produced by combining some biopolymer with a natural active agent. Incorporation of essential oils into starch-gelatin blend [60], alginate-clay [64], pullulan [63] or chitosan [62] films provided not just antifungal activity, but also improved film structural properties. Antifungal dermal patches have also been developed using more specialized polymeric matrices and most often one of clinical azoles [65,66]. So far, antifungal polyenes have not found their way in the film-based delivery. More recently, efficient chitosan–polyethylenimine copolymer microneedle patches loaded with AmB have been developed and have been demonstrated to be effective in a *C. albicans* infection mouse model [67]. In comparison, the mcl-PHA platform, described in the current study, is simpler, cost effective and could be applied for different antifungal and antibacterial agents, thus providing platform to combat bacterial biofilms. Solvent casting is a simple, straightforward procedure frequently employed for production of PHA-based biomaterials in the form of films. By adding different constituents into a film solution, or combining this technique with another one such as particle leaching, it is possible to modulate both functionality and structural properties of the obtained film samples. Rai et al. have developed polyhydroxyoctanoate (PHO)/bio-glass nanocomposite elastomeric films with a unique ability to support vascularization [8]. The presence of antimicrobial and bioactive bio-glass made films suitable for tissue engineering and wound dressing. The impregnation of mcl-PHA films with bioactive, inorganic constituents (e.g., hydroxyapatite [68] and titanium dioxide [69]) has also improved their mechanical properties and bioactivity, potentially leading to osteoinductive scaffolds. Porous and neat PHO films, designed by Bagdadi et al. as cardiac patches, supported viability, adhesion and contraction of neonatal ventricular rat myocytes [15]. Blends of P(3HO)/P(3HB) films were loaded previously with aspirin for biodegradable drug-eluting stents [70].

Biofilm-associated fungal infections mostly arise from cells that colonize the surfaces of implanted medical devices [71]. Due to high antifungal resistance of biofilms, these infections are very hard to treat and present an important clinical problem [72,73]. Therefore, preventing fungal growth and biofilm formation on medical devices is a promising strategy to combat these infections. So far, several approaches were used, such as “lock” therapy (incubating catheters in high concentration of an antimicrobial drug prior to their contact with the patient), coating the surfaces of medical devices with given molecules or modification of polymers incorporated in medical devices [25]. Antibiofilm properties of PHA film preparations containing polyenes indicate their potential in the context of medical devices and implants as coatings to prevent biofilm formation.

## 4. Materials and Methods

### 4.1. Mcl-PHA and Monomers Production

Mcl-PHA, more specifically, polyhydroxyoctanoate-co-decanoate (P3(HO-co-DO)) was produced with *Pseudomonas putida* KT2440 strain with octanoic acid in the fermentation feed as the sole energy and carbon source, extracted with ethyl acetate and characterized as in our previous manuscript [74]. This polymer was determined to contain an equal ratio of hydroxyoctanoate to hydroxydecanoate monomeric units and was further used to cast films. The polymer was also degraded to its monomeric units via an acidic methanolysis, followed by saponification with LiOH [18] to yield free hydroxy fatty acids (HAs)–a mixture of (*R*)-3-hydroxyoctanoic and (*R*)-3-hydroxydecanoic acids in molar ratio of 1:1. This mixture was separated by liquid column chromatography in acetonitrile gradient to yield purified single hydroxylated fatty acids [44].

### 4.2. Preparation of mcl-PHA/Polyene Films

Mcl-PHA/polyene films were prepared by the solvent casting method. A film solution was prepared by dissolving PHA and polyenes in an appropriate volume of chloroform at 25 °C, after which it was poured into a glass Petri dish (90 mm diameter) and left to dry in the air at ambient temperature for 4 days. The PHA content, as well as the mass of the incorporated polyenes was varied (Table 2).

In addition, mcl-PHA films, loaded with AmB/C10 monomer (sample AMB-M; Table 1), were prepared under the same conditions. The pure mcl-PHA film (PHA-control), without polyenes, was made for comparison.

### 4.3. Characterization of mcl-PHA/Polyene Films

#### 4.3.1. Scanning Electron Microscopy (SEM)

Analysis of samples surface and internal structure were performed on a Zeiss Ultra Plus microscope (Carl Zesis AG, Oberkochen, Germany). The samples were frozen under liquid nitrogen, fractured, coated with gold/palladium (50/50), and observed using an accelerating voltage of 5 kV.

#### 4.3.2. Differential Scanning Calorimetry (DSC) and Thermogravimetric Analysis (TGA)

DSC experiments were performed on a DSC 4000 apparatus (Perkin Elmer, Waltham, MA, USA). Mcl-PHA and polyene containing mcl-PHA films of about 4 mg were used. Samples were heated from −40 °C to 100 °C at a rate of 10 °C/min under nitrogen atmosphere and held for 2 min before cooling at 10 °C/min. A second heating scan was performed at 10 °C/min to characterize all materials after exactly the same thermal history.

TGA was performed on a Pyris 1 analyzer (Perkin Elmer, Waltham, MA, USA). The samples were heated from 30 to 600 °C at a rate of 10 °C/min under air flow.

#### 4.3.3. Water Contact Angle (WCA)

PHA and modified PHA samples were characterized by evaluating their contact angle using assembled goniometer set-up. The change in hydrophilicity and hydrophobicity of the PHA films was assessed by measuring the contact angle based on the static sessile drop method. Single water droplets of 10 µL were deposited on the PHA film surface using a gilmount syringe. The equilibrium water contact angle was measured as the mean of the right and left contact angles. The average of five readings of equilibrium water contact angles was calculated and reported as the water contact angle of samples.

#### 4.3.4. Release Kinetics

In vitro release studies of Nys and AmB from PHA-films were performed in simulated sweat [75] at 37 °C in a water bath with mechanical agitation (WND14 Memmert, Schwabach, Germany). Prepared films were cut into rectangles (3.5 × 1 cm^2^) and placed in 10 mL of simulated sweat at pH 5.4. At specific time intervals aliquots (3 mL) were withdrawn and simultaneously replaced with fresh 3 mL of simulated sweat. Two independent experiments were carried out.

Withdrawn samples were immediately filtered through Whatman filter paper and assayed spectrophotometrically for Nys and AmB content in triplicate by an UV/VIS spectrophotometer (Shimadzu 1800, Kyoto, Japan) at a maximum absorption wavelength, λ_max_, 305 and 416 nm, respectively [56,76,77], using a standard calibration curve.

### 4.4. In Vitro Antifungal Activity

Fungal strains used in this study belonged to genus *Candida* (*C. albicans* ATCC 10231, *C. albicans* SC5314 (ATCC MYA-2876), *C. parapsilosis* ATCC 22019) or to the group of filamentous fungi (*Aspergillus fumigatus* ATCC 13073; *Trichophyton mentagrophytes* ATCC 9533, *Microsporum gypseum* ATCC 24102). *Candida* spp. were routinely grown in Sabouraud media (SAB; 4% glucose, 1% Bacto peptone, pH 5.6, plus 2% agar for solid medium) at 37 °C. *Candida* cells were re-suspended in SAB medium to an OD_500_ of 0.1 and 200 µL of cell suspension was spread on SAB agar plates. Filamentous fungi were grown on Potato dextrose agar plates (PDA; Himedia, Mumbai, India) at 30 °C. Spores were re-suspended in sterile water supplemented with 0.1% Tween 20 and counted using a hemocytometer chamber. The spore number was adjusted to 10^5^ CFU/mL and 200 µL of cell suspension was spread on SAB agar plates. Films containing polyenes (6 mm diameter) were placed on the surface. The plates were incubated at 37 °C and growth inhibition diameters were measured after 24 and 48 h for *Candida* spp. and up to 7 days for filamentous fungi. Alternatively, rectangular shapes (1 × 1 cm^2^ and 1 × 2 cm^2^) and incubation temperature of 30 °C were also tested. Antifungal activity of the films was also tested in liquid SAB medium inoculated with 10^2^ fungal cells per well (6-well plate). Fungal growth was monitored daily and compared to the control (fungal cells in media, without films). Two independent experiments with triplicates per test were carried out.

The minimum inhibitory concentration (MIC) of amphotericin B, nystatin and PHA monomers (C8-OH and C10-OH) were determined according to standard EUCAST method for susceptibility testing of yeasts (E.DEF 7.39) and molds (version 9.2) using the concentration range 500–0.06 µg/mL. Fractional inhibitory concentration index (FICI) between mcl-PHA monomers and nystatin and AmB were determined using *C. albicans* ATCC 10231 strain and interpreted (synergy -FICI ≤ 0.5; antagonism -FICI > 4.0 and no interaction -FICI > 0.5–4) using the previously described checkerboard microdilution protocol [78].

### 4.5. Candida Biofilm Inhibition Test

A biofilm inhibition test was performed using the clinical biofilm producer *Candida albicans* 329UCK, obtained from the Centre for Preclinical Research, Medical University of Warsaw. The clinical strain was stored in SAB medium supplemented with 20% glycerol at −80 °C. Prior to experiments, clinical isolate was sub-cultured onto Saboraud dextrose agar (SDA) and incubated at 30 °C for 48 h. In order to standardize cell suspension, 20 mL of RMPI-1640 (SIGMA, St. Louis, MO, USA) supplemented with 2% wt glucose and buffered with MOPS (POL-AURA) was cultured with colony of *Candida* strain and incubated overnight in a rotary shaker (100 rpm) at 35 °C. The cells were harvested by centrifugation, washed twice in sterile deionized water, re-suspended in RMPI medium and counted in hemocytometer to obtain 1–5 × 10^6^ cells/mL. Three independent experiments were carried out.

In the first experimental series, antifungal activity of the sample films against the clinical *Candida* isolate was determined using agar cup method. Mueller–Hinton agar supplemented with 2% glucose and methylene blue (MH-GM) was inoculated by spreading the *Candida* spp. (1 × 10^6^ CFU/mL) isolate over the entire agar surface. Following agar solidification, a sterile cork borer was used to punch holes (5 mm in diameter) and samples films (round pieces, diameter 5 mm) where placed into the obtained wells and filled with SAB. The plates were incubated at 35 °C and growth inhibition diameters were measured after 24 h.

For the biofilm inhibition evaluation, the round film samples (10 mm in diameter) attached to a circular microscope slide with a double-sided tape were used. In order to allow adhesion, *Candida* cells (5 × 10^6^ cells/mL) were incubated in the presence of films for 180 min at 35 °C, under agitation (100 rpm). After incubation, glasses coated with films were rinsed with saline and transferred into 24-well plate filled with fresh RPMI medium. To allow biofilm formation, the plates were incubated at 35 °C for 48 h under agitation (100 rpm). The medium was replaced after 24 h. Following incubation, all samples were removed and washed to remove non-adherent cells. Samples with a formed biofilm were transferred into a 5 mL tube filled with saline and 2 mm glass beads and then homogenized (MP Biomedicals). Biofilm biomass was collected by centrifugation, sonicated and then pelleted by centrifugation. The obtained cell pellets were re-suspended in fresh saline and aggregates were dissolved by de-gas followed by sonication. Cell dilutions in saline were prepared and plated on SDA plates. Following overnight incubation (35 °C), growth colonies were counted and number of colony-forming units (CFU) per slide was the determined.

A replica of each attached film sample was used for microscope analysis. The colonized surface was visualized by staining with Crystal Violet (1%, *w*/*v*) and standard microscope (NIKON Eclipse 50i and 400× magnifications).

To investigate the potential of the obtained films in biofilm eradication, a filter biofilm model was employed [79] with some modifications. Micropore filters (Millipore) were placed on top of an SDA plate. *Candida* inoculum (5 × 10^6^ cell/mL, 5 µL) was placed on top of the filter and incubated at 35 °C for 48 h. For biofilm to develop, inoculated filters were transferred to a fresh SDA plate every 24 h. Following biofilm formation, the PHA/polyene films were placed on top of each *Candida* biofilm. The sample films were in the form of a dressing, sandwiched in between two pieces of gauze previously soaked into buffer for full hydration. All treatments were incubated for another 24 h in high humidity conditions. The biofilm eradication effect was evaluated by streaking from the filters and replication imprints of the filters and PHA-based films, following the treatment. The final results were evaluated by presence or absence of *Candida* growth, after the treatment when sub-cultured and replica-plated onto a fresh SAD.

### 4.6. Cytotoxicity Test

Cytotoxicity of polyenes and monomers was determined as antiproliferative activity by standard (3-(4,5-dimethylthiazol-2-yl)-2,5-diphenyltetrazolium bromide (MTT) assay on human lung fibroblasts (MRC5; ATCC collection) and expressed as the concentration of the compound inhibiting cell growth by 50% (IC_50_) [80]. Compounds were tested in a concentration range (50–1 µg/mL for polyenes and 2000–25 µg/mL for PHA monomers) with 5 different concentrations used and each concentration was tested in quadruplicate.

### 4.7. Statistical Analysis

Statistical analysis was done using Student’s t-test and one-way ANOVA. The results are presented as mean ± standard deviations (SD). The difference was considered to be statistically significant at *p* ≤ 0.05.

## 5. Conclusions

New mcl-PHA/antifungal polyene film formulations with Nys and AmB were synthesized by the solvent casting method. Incorporation of the polyenes did not significantly affect thermal properties and the stability of the obtained films; however, it increased their hydrophobicity. All formulations exhibited excellent antifungal activity against yeasts and filamentous fungi in vitro. The addition of the PHA monomer, 3-hydroxydecanoic acid (C10-OH,) to the film solution has improved the overall antifungal activity due to the synergistic effect with polyenes. All antifungal PHA films prevented the formation of *C. albicans* biofilm, while they were not efficient in the eradication of the formed biofilms, the obtained results suggest that mcl-PHA-based polyene films could potentially be used as antifungal coatings for medical implants and devices.

## Figures and Tables

**Figure 1 antibiotics-10-00737-f001:**
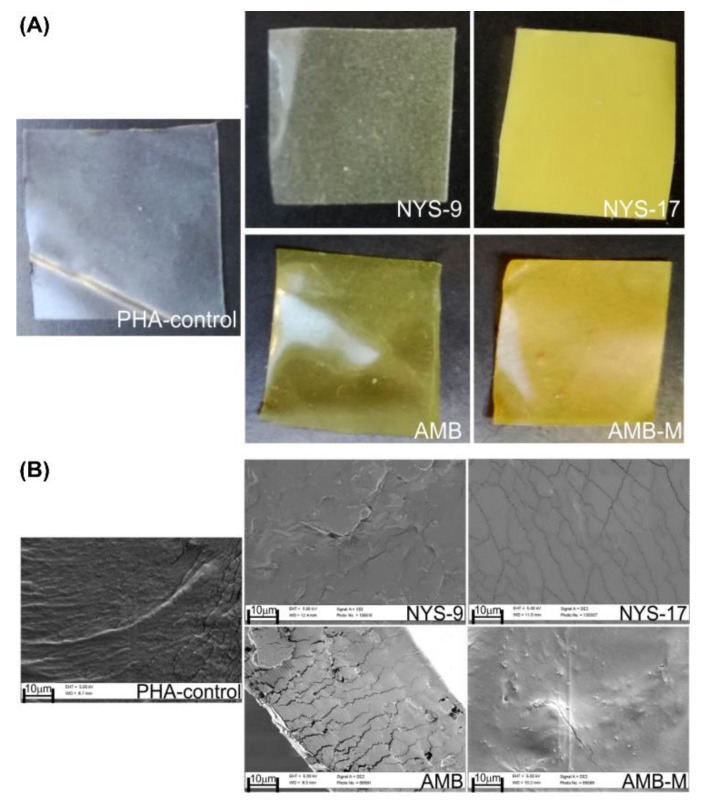
(**A**) Photographs and (**B**) SEM micrographs of the mcl-PHA films (neat and loaded with NYS-nystatin and AMB-amphotericin B). AMB-M sample contains 0.4 % wt C10-OH monomer.

**Figure 2 antibiotics-10-00737-f002:**
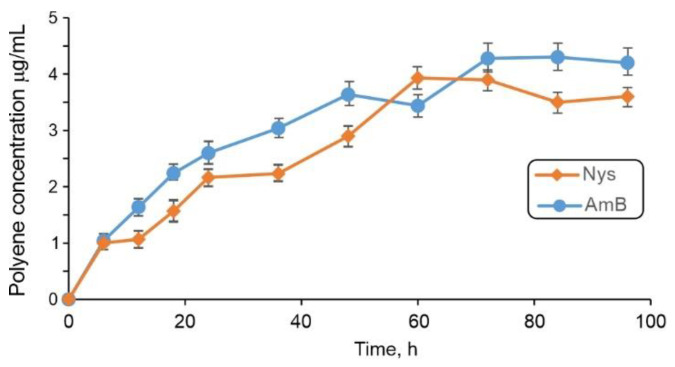
Release of polyenes (Nys and AmB) in simulated sweat from mcl-PHA films NYS-9 and AMB over 4 days at 37 °C. Values are means of two independent experiments ± standard deviations (SD).

**Figure 3 antibiotics-10-00737-f003:**
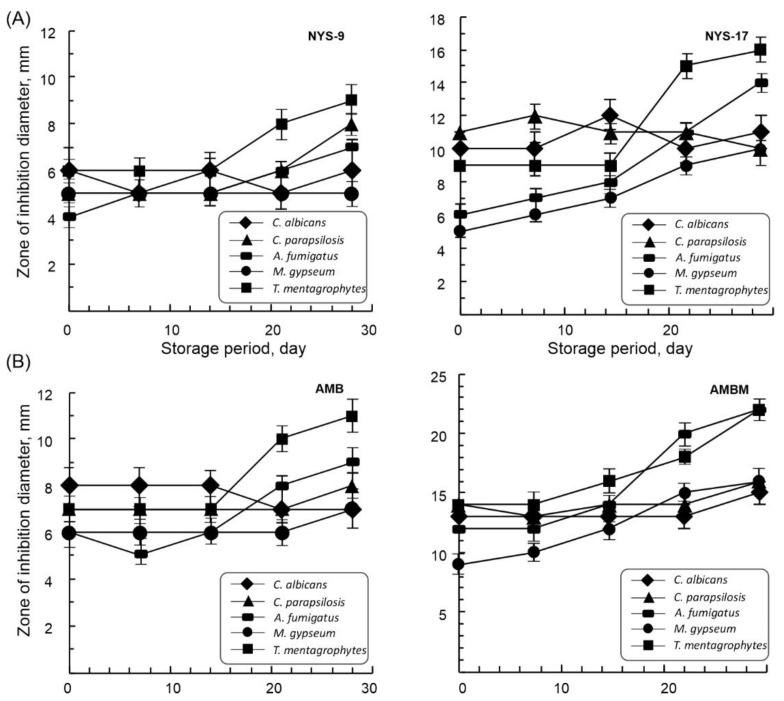
Zones of inhibition and stability of mcl-PHA films containing polyenes; (**A**) NYS (NYS-9, NYS-17 with 0.2% and 2% wt of nystatin, respectively), and (**B**) AMB (AMB and AMB-M with 0.1% and 1% wt of amphotericin B and 0.4% wt C10-OH monomer, respectively).

**Figure 4 antibiotics-10-00737-f004:**
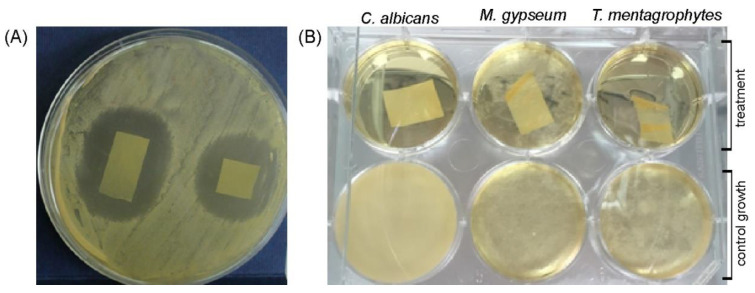
In vitro antifungal activity of AMB-M film (**A**) in direct contact on agar plates against *C. albicans* after 24 h incubation at 37 °C and (**B**) in broth assay with *C. albicans*, *M. gypseum* and *T. mentagrophytes* (control fungal growth in SAB medium is shown in bottom wells).

**Figure 5 antibiotics-10-00737-f005:**
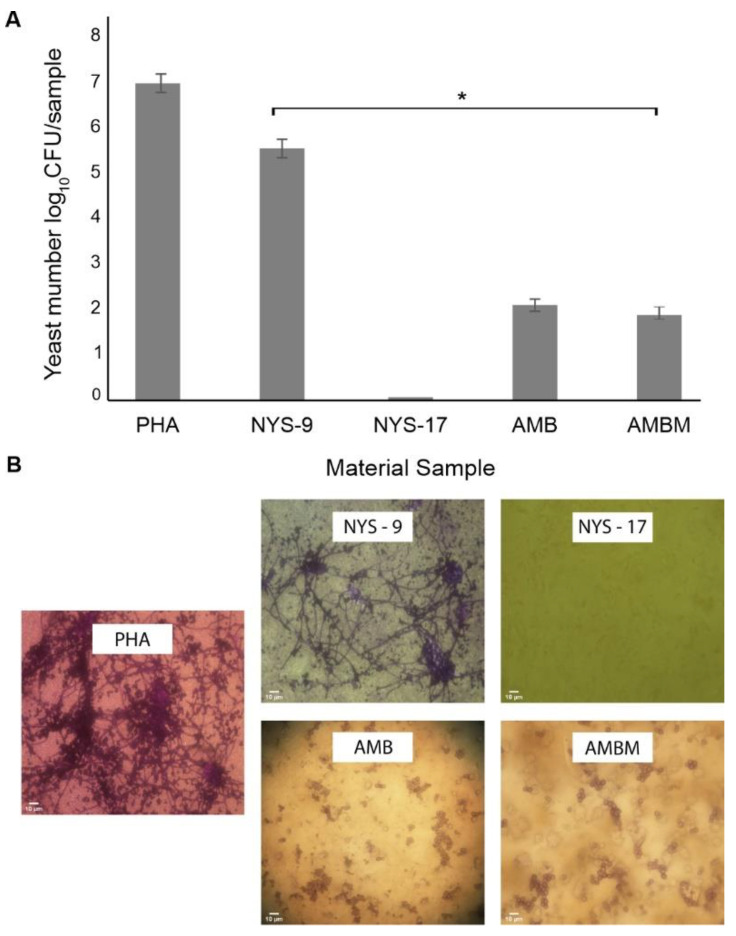
Biofilm formation of *Candida albicans* 329UCK grown in the presence of mcl-PHA films. (**A**) Adhesion of *C. albicans* to mcl-PHA films expressed as colony forming units per sample. The data are expressed as the mean ± SD of three separate experiments (* *p* ≤ 0.05). (**B**) Microscopic images of materials containing *C. albicans* biofilms. Scale bar,10 µm. NIKON Eclipse 50i (400× magnifications).

**Figure 6 antibiotics-10-00737-f006:**
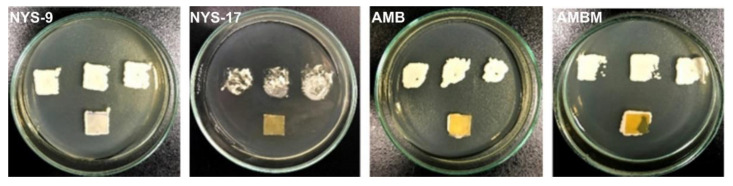
The replication imprints of PHA-polyene films, after 24 h treatment of mature *Candida albicans* biofilms grown on a micropore filter.

**Table 1 antibiotics-10-00737-t001:** MIC values (µg/mL) of polyenes, mcl-PHA monomers (C8-OH and C10-OH) and combination of C10-OH/AmB and their cytotoxicity values (IC_50_, µg/mL) against healthy human fibroblast cell line (MRC-5).

Compound	*Candida Albicans*	*Candida Parapsilosis*	*Aspergillus* *Fumigatus*	*Microsporum Gypseum*	*Trichophyton Mentagrophytes*	MRC-5
Nys	1.00	1.00	4.00	4.00	8.00	25.00
AmB	0.25	0.25	2.00	2.00	0.50	4.00
C8-OH	500	>500	>500	250	500	1000
C10-OH	250	250	500	250	250	800
C10-OH/AmB	6.25/0.125	15.60/0.125	6.25/1	15.60/1	/ ^1^	500–800

^1^ No synergy was detected with sub-inhibitory concentrations.

**Table 2 antibiotics-10-00737-t002:** Film samples, codes and compositions prepared via solvent casting approach.

Sample	mcl-PHA [g]	Nys [g]	AmB [g]	CHCl_3_ [mL]	% wt Polyene per Film Solution
**PHA**	1.0	-	-	6	-
**PHA/Nys**					
**1**	0.50	0.01	-	6.00	0.10
**2**	1.00	0.02	-	12.00	0.10
**3**	2.00	0.20	-	12.00	0.10
**4** (**NYS-9**)	0.50	0.01	-	3.00	0.20
**5**	2.00	0.30	-	12.00	1.50
**6** (**NYS-17**)	1.00	0.20	-	6.00	2.00
**7**	1.00	0.20	-	3.00	3.50
**PHA/AmB**					
**8** (**AMB**)	0.50	-	0.006	2.50	0.10
**9**	0.40	-	0.04	2.50	1.00
**10** (**AMB-M**) **^1^**	0.40	-	0.04	2.50	1.00

^1^ AMB-M = Amphothericin B and PHA monomer containing C10-OH monomer with AmB/monomer ratio 25:1.

## Data Availability

The data presented in this study are available on request from the corresponding author.

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
