# Peer review of "Polyhydroxyalkanoate/Antifungal Polyene Formulations with Monomeric Hydroxyalkanoic Acids for Improved Antifungal Efficiency"

_antibiotics, 2021, doi:10.3390/antibiotics10060737_

Round 1
Reviewer 1 Report
In this work Pekmezovic et al described antifungal properties of Mcl-PHA based polyene formulations. The paper is professionally written, the data presentation is clear, and the methods are adequately described. Minor comments are provided below.
Lines 8-20: Please provide consistent layout of affiliations (eg. Institute/Department, Faculty, University etc.)
Please change in-vivo/in-vitro to in vivo/in vitro within whole manuscript.
Line 108: please provide expansion of the abbreviation: C8-OH and C10-OH (as it is made for C8-OH in Discussion: line 240, while in line 243 it is not provided for C10-OH).
Line 127: please standardize descriptions of Tables and Figures within manuscript (sentence is finished with “.” in Table 2, while in Table 1 is not).
Authors should include Figure 1 after e.g., lines 140-153.
Please standardize charts among whole manuscript (e.g., chart on Figure 2 is provided without SD, while on Figure 3 it is extensive).
Line 189: Please provide full name of microorganism when it is appearing for the first time within the manuscript e.g., Candida albicans (full name is provided in line 201).
Line 223: C. albicans in italics.
Line 285: et al. not at el.
Line 380: Please standardize the reference.
Reference section: please standardize whole reference section (once doi is included, once not etc.)
Reviewer 2 Report
In this manuscript authors used medium chain length polyhydroxyalkanoates (mcl-PHA) to formulate both polyenes (Nys and AmB) in a form of films. Thus, thermal properties and stability of the materials were not significantly altered by the incorporation of polyenes in mcl-PHA. Mcl-PHA based polyene. The formulations showed high growth inhibitory activity against yeasts (C. albicans ATCC 1023, C. albicans SC5314 (ATCC MYA-2876), C. parapsilosis ATCC 22019) and filamentous fungi (Aspergillus fumigatus ATCC 13073; Trichophyton mentagrophytes ATCC 9533, Microsporum gypseum ATCC 24102). Furthermore, the antifungal PHA film prevented the formation of C. albicans biofilm rendering them suitable for the transdermal or as implants coating agents.
This manuscript presents significant results and is of great importance for researchers in scientific and medical fields with potential industrial applications. Besides, this manuscript is well sectioned and properly written. Nevertheless, some comments still needs to be addressed. In summary, this paper is suitable for publication in antibiotics after minor modifications.
comments:
- Title : very clumsy and complex…needs to be rewritten
- Line 37: … replace there with they
- Was the formulated polymer tested against bacterial biofilms to see if it may have a common effect on all type of biofilms or it was just tested with yeast and fungal biofilms? Thus, it may be written as a perspective.
- Fig 2 the error bar should be added
- I see no description or hypothesis about how this polymer is acting to combat biofilm attachment to the surface …maybe adding a figure to illustrate may be beneficial.
- What type of statistical tests were used to compare the results? I see no indications about statistical analysis and the P value especially when comparing the results presented in Fig 2, 3 and 5.
- How many times were the tests repeated? This should be clearly mentioned for all the conducted tests (each one alone).
- A conclusion section of this work should be added that may include the main findings and the perspectives of this discovery including industrial applications and further testing.
Reviewer 3 Report
Over all the research very intresting and well organiezed,well written.how ever,after my evalution still the paper need some improvements. other comments appended the below.
1.what is the difference from (C. albicans ATCC 1023, C. albicans SC5314 (ATCC MYA-2876
2.add some points in source of Polyhydroxyalkanoates (PHAs)
3.what is the tensile strength and viscocity of the films??
4,mention the culture incubation time of all fungi
5.IC50 change to IC50
6,some of the place C.albicans not italics,revise this all places
7.avoide the typographical and syntx errors
8.microscopic images mention the what type of microscope used??
9. add the methodology of microscopic observation of fungal biofilms??
10.what is the thickness of biofilm control and treatments??
11.authour why not study the Confocal laser scanning microscopical observation of fungal biofilm??
12.authour add the following statistical analysis,its missing the manuscript
13.add the significant asterrisks in bar diagram images
14.how many replications or triplications used??
15.what is the positive and negative control used for this experiments??
16.authour add the cytotoxicity microscopical image of lung fibroblast cells
Reviewer 4 Report
The article by Pekmezovic et al describes the usefulness of a modified formulation of polyenes in treating variety of fungal infections. The article is well written. The significant of the study needs to be reflected in a better way in the introduction. Here are my comments
- What are the differences between PHA/PHB based modifications of polyenes compared to the
different modified polyenes described previously in https://www.mdpi.com/2079-6382/9/6/312/htm
https://pubmed.ncbi.nlm.nih.gov/23729001/ https://www.ncbi.nlm.nih.gov/pmc/articles/PMC4856207/ Please clarify. The authors mention about lipid formulations in lines 82-89 but should explain for better clarity how PHA/PHB based modifications may be more advantageous to the already present modifications. This will enhance the significance of the article. - Lines 117-120: AMB MICs were decreased only 2-fold. A significant change in MICs is considered equal to or greater than 4-fold. Please clarify? Is the subtle change in MIC worth the modification proposed?
- Did the authors test any polyene resistant strains from Candida or Aspergillus or others for synergy and lowering of resistance ? If not it will make the paper stronger if the authors can show a decrease in polyene resistance in strains when modified with this compound in some polyene resistant strains.
- Table 1: How does these modifications compare to the synergistic effects of the previously modified AMB as described in https://www.mdpi.com/2079-6382/9/6/312/htm, https://pubmed.ncbi.nlm.nih.gov/23729001/. https://www.ncbi.nlm.nih.gov/pmc/articles/PMC4856207/ ? Please explain
- Fig 2: Since it is mean of two experiments, the authors should put the error bars in the figure
